# Expectations for the Development of Health Technology Assessment in Brazil

**DOI:** 10.3390/ijerph182211912

**Published:** 2021-11-13

**Authors:** Julia Simões Corrêa Galendi, Carlos Antonio Caramori, Clarissa Lemmen, Dirk Müller, Stephanie Stock

**Affiliations:** 1Institute for Health Economics and Clinical Epidemiology (IGKE), Faculty of Medicine and University Hospital Cologne, University of Cologne, 50935 Cologne, Germany; clarissa.lemmen@uk-koeln.de (C.L.); dirk.mueller@uk-koeln.de (D.M.); stephanie.stock@uk-koeln.de (S.S.); 2Department of Internal Medicine, Medical School, São Paulo State University (UNESP), 18618687 Botucatu, Brazil; carlos.caramori@unesp.br

**Keywords:** health services research, surveys and questionnaires, health technology assessment, decision making, redesign, Brazil

## Abstract

The implementation of health technology assessment (HTA) in emerging countries depends on the characteristics of the health care system and the needs of public health care. The objective of this survey was to investigate experts’ expectations for the development of HTA in Brazil and to derive measures to strengthen the impact of HTA in Brazil on health care decisions. Based on a scoping literature review, a questionnaire was developed proposing eight theses for seven domains of HTA: (i) capacity building, (ii) public involvement, (iii) role of cost-effectiveness analysis (CEA), (iv) institutional framework, (v) scope of HTA studies, (vi) methodology of HTA, and (vii) HTA as the basis for jurisdiction. Thirty experts responded in full to the survey and agreed to five of the eight theses proposed. Experts suggested several measures to promote HTA within the scope of each domain, thus addressing capacity building related to HTA, availability, and reliability of population data, and legal endowment of the HTA system. Finally, HTA processes in Brazil should also address public health issues (e.g., appraisal of interventions directed at chronic diseases).

## 1. Introduction

Health technology assessment (HTA) is a multidisciplinary process that aims to systematically assess the value of a health technology including, but not limited to, clinical, epidemiological, and health economic evidence [1]. The increase in medical expenditures and the limited resources available in most health systems have claimed for decisions to be guided by evidence and sound rationales [2]. HTA has been an important input to health policy formulation and implementation in different settings, especially when applied to inform decision making with regard to coverage and pricing of new medical technologies and to promote best clinical practices [3].

The trajectories and organizational structures of HTA have focused on the assessment of evidence on drugs and clinical issues, whereas it can also be applied to public health interventions such as the prevention and treatment of chronic diseases [4]. In low- and middle-income countries, chronic diseases represent a growing burden, accounting for more than 80 percent of global cardiovascular, respiratory, and related chronic disorder disability-adjusted life years (DALYs) and deaths, respectively [5]. In light of unprecedented technological innovation, population aging, and economic concerns in emerging countries, HTA has to consider (i) how to prioritize interventions that are aimed at preventing, diagnosing, and treating chronic diseases, and (ii) how to account for the social, ethical and legal implications of increasingly expensive and complex interventions [6]. Thus, the implementation of an HTA body may be an essential pillar for a growing national health system.

The implementation of HTA depends on characteristics of the health care system, cultural and social preferences, availability of resources, and institutional capacity [7]. However, there are many overarching efforts that encompass region-wide HTA initiatives in Latin America, and stand-out experiences from selected countries on how to implement HTA at a national level [8]. Latin America faces significant challenges in consolidating the influence of HTA on decision making and tackling the right-to-health litigation issue, the latter being a clear consequence of the perceived inadequate access to health services [8].

As the largest country in Latin America, Brazil can be considered exemplary for emerging countries. Health care in Brazil is guaranteed by a decentralized, universal public health system (SUS), which is funded by tax revenues and contributions from federal, state, and municipal governments [9]. The implementation of HTA in Brazil has been advancing fast since the Policy of Health Technology Management, enacted in 2009, when a centralized structure was implemented to inform the decision-making process [10].

By inquiring Brazilian experts from different fields, the objective of this survey was (i) to anticipate the development of HTA in Brazil for the next 10 years and (ii) to identify potential measures to stimulate HTA in Brazil.

## 2. Materials and Methods

To guide the expansion of HTA, it is important to consider different perspectives on the potentialities and the current challenges [7]. Hence, an e-survey with Brazilian experts was conducted from 31 May to 13 June 2019 using an online tool (Survey Monkey Inc., San Mateo, CA, USA).

### 2.1. Development of the Questionnaire

A scoping literature review was performed in Medline and Virtual Health Library (BIREME) to identify potential barriers for the HTA implementation in Brazil and Latin America. For the scoping review, an iterative process similar to that of Arksey and Malley was followed: (i) identifying the research question, (ii) identifying relevant studies, (iii) study selection, (iv) charting the data, and (v) consultation exercise with two experts [11]. Two experts who did not participate in the survey provided additional references about potential studies to include in the review, as well as valuable insights about issues relating to HTA in Brazil. Data charting was performed based on a standard form that included the publication identifiers (i.e., author, year, and title), setting, design, main objective, and key findings [11]. Eligible were publications addressing the Brazilian health policies for the establishment of an HTA system, assessments of institutions, or of HTA reports in Brazil. The literature search comprised the time span from January 2009 to January 2019 (i.e., because the current structure for HTA was established in Brazil in 2009). Articles in Portuguese, Spanish, and English were included. Grey literature was searched by revising the citations of included articles and reports from conferences focused on HTA (i.e., Latin American Policy Forum from HTAi) and websites of relevant institutions (i.e., CONITEC, Brazil; IECS, Colombia; Pan American Health Organization (PAHO)). The search strategy comprised the terms “Health technology Assessment”, “Brazil”, and “Latin America” and can be found in full on the Appendix A.

A total of 29 studies identified through the literature review guided the development of the questionnaire [8,10,12,13,14,15,16,17,18,19,20,21,22,23,24,25,26,27,28,29,30,31,32,33,34,35,36,37,38]. Six studies addressed the right-to-health litigation issue in Brazil [12,13,14,23,28,37], five studies discussed the role of cost-effectiveness analysis in decision making in Brazil [29,30,32,33,35], three studies addressed the public involvement on HTA processes in Brazil [17,21,34]. In addition, 15 studies offered a comprehensive assessment of HTA in Brazil. Specifically, two studies were based on surveys with Brazilian experts about the HTA implementation in Brazil [19,31], five studies were opinion pieces presenting a critical appraisal on the current HTA institutions in Brazil [8,10,24,25,36], seven were reviews of the Brazilian HTA reports [15,16,18,20,26,27,38], and one study was a literature review addressing the use of real-world data on Latin America [22].

Based on the barriers identified in the literature review, the questionnaire addressed seven domains: (i) capacity building (i.e., how individuals achieve skills, experience, and knowledge to interpret and conduct HTA studies) [39], (ii) public involvement, (iii) role of cost-effectiveness analysis (CEA), (iv) institutional framework, (v) scope of HTA studies, (vi) methodology of HTA, and (vii) HTA as the basis for jurisdiction.

For each domain, the questionnaire was structured, as shown in Figure 1. First, a thesis projecting a possible situation was stated. Then, the experts should rate how possible the proposed thesis in a 10-year time frame is (from very possible to very impossible). Second, measures to address each domain were stated, and experts should rate them from strongly disagree to strongly agree. Lastly, open-ended questions allowed participants to justify their answers, as well as to suggest other measures not provided in the questionnaire.

All responses were rated on a 4-point Likert scale with no option of a neutral response. The questionnaire contained one thesis for each of the domains: “Education in HTA”, “public involvement”, “role of CEA”, “scope of HTA studies”, “methodology of HTA” and “HTA as basis for jurisdiction”. For “institutional framework”, two theses were proposed. The rationale behind each of the eight theses is outlined in Appendix A, and the full version of the questionnaire is provided.

The questionnaire was written in Portuguese language and was pre-tested for comprehensibility with three Brazilian academics who did not participate in the survey. Results of the survey are reported in accordance with the CHERRIES checklist for reporting results of e-surveys [40].

### 2.2. Selection of Participants

In Brazil, a centralized HTA structure was created to assist the Ministry of Health. The National Commission for Incorporation of Technologies (CONITEC) operates mainly by issuing reports to inform coverage decisions, and both methodological guidelines and national disease-management guidelines to support integral health care [41]. In addition to the centralized HTA process in Brazil, the Nuclei of Health Technology Assessment (NATS) were established in high complexity hospitals and universities. The NATS compose a network of hospital-based HTA, whose aim is to promote the HTA rationale across the country [41].

Participants were selected mainly through purposive sampling [42]. Invitations were sent to all members of the plenary of CONITEC, employees of the Department of Science and Technology (DECIT) of the Ministry of Health, and coordinators from NATS. Experts of the pharmaceutical industry were invited if they had at least 10 years of experience in HTA or had published relevant articles on HTA. The invited experts were informed about the aim of the study and the subsequent data processing (i.e., anonymous participation and data privacy). The experts who agreed to participate signed informed consent and received the link to the survey by e-mail. Reminders were sent out every two weeks twice if necessary.

### 2.3. Data Analysis

The data of Likert scales were analyzed by descriptive statistics as mean, standard deviation (SD), mode, and frequencies. A calculation was performed by attributing a code to each alternative: strongly agree (very likely): 4, agree (likely): 3, disagree (unlikely): 2, and strongly disagree (very unlikely): 1. Text generated in open-ended questions were analyzed step-by-step by reading thorough all responses in order to become familiar with the data, followed by identifying codes and themes. Finally, themes were reviewed and mapped according to recurrence [43].

## 3. Results

In total, 36 responses were submitted from 31 May to 13 June 2019 (response rate: 29 percent). Thirty complete responses were included in the final analysis (83%). Overall, 63% of participants were representatives of NATS, 16% of regulatory agencies (i.e., CONITEC and DECIT), and 20% of pharmaceutical industries. Table 1 describes the participants according to their background.

### 3.1. Agreement to the Theses

At least 70% (mean ≥2.7) of the experts estimated that five of the eight theses will be developed in ten years (i.e., for the domains “capacity building”, “role of CEA”, “scope of HTA studies”, “methodology of HTA”, and “HTA as basis for jurisdiction”). Lack of funding and the current insufficient technical capacity in Brazil were most frequently cited as challenges to the achievement of the projections. Table 2 comprises the expectations of experts on how possible the proposed theses are and what challenges may impact their achievement.

At least 70% (mean ≥2.7) of the experts estimated that five of the eight theses will be developed in ten years (i.e., for the domains “capacity building”, “role of CEA”, “scope of HTA studies”, “methodology of HTA”, and “HTA as basis for jurisdiction”). Lack of funding and the current insufficient technical capacity in Brazil were most frequently cited as challenges to the achievement of the projections. Table 2 comprises the expectations of experts on how possible the proposed theses are and what challenges may impact their achievement.

Experts’ expectations differed in three of the eight theses (i.e., for the domains “public involvement” and “institutional framework”). According to the experts, there are two challenges for increasing public participation. First, in Brazil, patient representatives or collaborative networks exist for a few groups of diseases (cited by six experts). Second, conflict of interest could compromise the technical quality of the HTA process because of financial support from the industry to patient advocates (cited by eight experts). For “institutional framework”, experts perceive that one major barrier to merging private and public HTA appraisals is that private and public insurances in Brazil operate from different perspectives and, therefore, have different interests (cited by seven experts).

### 3.2. Measures to Enhance HTA in Brazil

Experts suggested several measures to strengthen HTA in Brazil, which are summarized according to the respective domains (i.e., i–vii). Table 3 shows in full the measures, upon which at least 70% of experts agreed or strongly agreed (mean ≥2.7), and measures suggested as free text.

i.Capacity Building

Experts agreed to several types of educational programs (i.e., post- and graduate programs, project-based programs, online and in-class traditional courses) and on institutions potentially responsible for their implementation (i.e., public universities or institutions, international collaborations, private universities, or public–private partnerships). It was recurrent in our sample that students frequently lack basic skills (i.e., evidence-based medicine, English language), which are not the focus of postgraduate programs.

ii.Public Involvement

Measures to enhance public participation include advertising the current mechanism of public consultation (i.e., the appraisal of CONITEC is currently made available online for a short period of time when any stakeholder of the society can submit comments or counterarguments. After the public consultation is closed, the contributions are evaluated by the plenary before issuing the final report). In addition, promoting both informative and educative interventions directed to patients’ representatives should improve the quality of these contributions.

iii.Role of Cost-Effectiveness Analysis (CEA)

More than 70% of experts agreed that a CET should be used as a reference for the interpretation of the results of the CEA, and a CET should be based on a standard value to be developed considering budget availability. However, a CET should be applied as one criterion among others (such as ethical, evidence of medical benefit, health care priority, etc.).

iv.Institutional Framework

Experts agreed that expanding capacities of the NATS and the CONITEC would fasten the timeframe of HTA processes. Experts also agreed to increase the participation of the National Regulatory Agency for Private Health Insurance (ANS for its acronym in Portuguese) in the plenaries of CONITEC.

v.Scope and vi. Methodology of HTA

To address the domain “scope of HTA”, a simplified appraisal procedure could be applied to medical technologies with low budget impact. Moreover, experts suggested that HTA should prioritize technologies aimed at primary health care. For “Methodology of HTA”, experts agreed to five measures that aimed at improving data availability and promoting technical capacity (i.e., training of researchers, financial support to independent researchers from public and private universities, improvement of data collection, and amelioration of the accessibility of DATASUS and investment on integrated electronic databases).

vi.HTA as the Basis for Jurisdiction

Experts agreed upon three measures to address this domain. First, to establish expert groups to advise the law courts, and to expand initiatives such as the NAT JUS (i.e., nuclei of Health Technology Assessment that were established exclusively to support the Judiciary regarding lawsuits concerning health coverage). Second, to implement an early awareness system (or horizon scanning) for identifying innovative medical technologies with the potential to become a target of lawsuits. Third, to bind an HTA appraisal by CONITEC to authorization to enter the market—which in Brazil is issued by the Brazilian Health Regulatory Agency (ANVISA for its acronym in Portuguese)—to ensure timely reports.

## 4. Discussion

In addition to international recommendations for good practice on HTA in Latin America [7,44,45,46], to guide the implementation or expansion of HTA processes, the specific national context (i.e., the state of HTA development in each country, the resources available, and the characteristics of the health care system and decision-making process) have to be considered [7]. Hence, this paper explored how the development of HTA in Brazil is perceived by a sample of national experts.

A questionnaire was built upon seven domains that reflect current discussions on HTA in Brazil. Experts agreed to five of the eight theses proposed but differed with regard to the three theses proposed for the domains “public involvement” and “institutional framework”. Notably, the promotion of functional capacities (i.e., methodological expertise and timeliness of reports) was considered essential for an HTA body.

For most low and middle-income countries, capacity building related to HTA is a major challenge, which hinders HTA on many levels [35,38,47]. For instance, it has been previously reported that a shortage of qualified personnel in Brazil compromises the methodological quality of HTA reports, resulting in a low number of reports finished in time [48]. Results of our survey suggest that, in the view of experts, capacity building stands out as a challenge, preventing the application of CEAs for coverage decisions and the enforcement of HTA as the basis for jurisdiction. Nevertheless, international initiatives and cooperation that are aimed at capacity building have shown to be able to mitigate these hazards (e.g., by stimulating collaboration between stakeholders and international partners) [48,49].

Experiences from developed countries, which count on a more structured HTA system, anticipate that HTA is still focused on tackling “one-dimensional” health issues, while health problems with higher complexity are omitted. As shown for France, The Netherlands, Sweden, and the United Kingdom, HTA assessments are often limited to a single technology, which can be defined with reasonable clarity, suitable studies/data are available or at least can be commissioned easily and is of immediate relevance [4]. However, in the long term, systematic evaluations of health programs for chronic diseases may reduce mortality and morbidity on a larger scale, thus saving costs for the health care system [50]. In line with this rationale, some experts suggested prioritizing technologies for the primary health sector.

Due to the increased complexity of health care resulting from chronic diseases, the impact of HTA on policy, organization, community, and individual levels should be systematically assessed. These assessments should include effects on several intermediate outcomes, and how these effects are mediated through multiple causal pathways. Due to its complex nature, risk-reducing strategies (e.g., the development of vaccines or genetic-based screening for breast cancer) and treatments (e.g., multidisciplinary care for chronic diseases such as diabetes) require sufficient evidence as well as a conceptual openness for methodological pluralism [6]. By including social, ethical, and economic issues, HTA may be a valuable tool for emerging countries for the improvement of health care for chronic diseases.

Notwithstanding, in emerging countries such as Brazil, the expansion of the HTA structure to address public health issues may be hampered by financial constraints and a lack of population data [22,51]. While funding remains a challenge, measures that aim at improving the availability and reliability of data were elicited in our survey (i.e., improving data collection and accessibility of DATASUS, implementation of electronic integrated databases in hospitals and primary health care) and may contribute to improvements of HTA reports. Currently, Latin American countries heavily depend on evidence from other countries to produce HTA reports and guide decision making in healthcare [31]. However, the transferability of results may be limited and misguide decision making [52].

Moreover, although not addressed in our survey, there is a growing interest in patient aspects of health policy research and consumer expectations in developed countries. However, most HTA agencies neither use and nor invest in scientific methods to generate knowledge and evidence about the patient’s perception of a technology [53]. For newly developed HTA structures such as those to be developed for Brazil, it would be useful to include these patient-centered elements immediately.

A previous survey conducted in 2015 discussed the preferred status of HTA in 10 years in Brazil, among other Latin American countries [31]. Nevertheless, our survey was directed to the Brazilian health care system and included other domains considered to be relevant to Brazil. Moreover, our survey also identified several subjective aspects that may influence HTA in Brazil in a subtle manner such as the mistrust between the judiciary and its technical advisors or, pharmaceutical companies could corrupt and take public participation to their advantage.

Several limitations of our survey have to be addressed. First, the low response rate, which might have resulted from the length of the questionnaire. Second, our sample of experts cannot be considered representative for all groups of stakeholders involved in HTA decisions, i.e., agents from regulatory institutions and the industry were underrepresented, while patients were not involved at all. The patients’ perceptions of these domains should be investigated in future research. Lastly, the application of Likert scales does not allow inferring preferences of experts among the measures proposed.

This survey reflects the expectations of a selected group of Brazilian stakeholders that can prompt further discussion in Brazil and other Latin American countries. As different stakeholders usually have different perceptions of the health care system and its needs, HTA systems have to show considerations for these perspectives [54]. By providing the perspectives of Brazilian stakeholders in a structured manner with regard to specific domains, the results of this survey can support health policies aiming at promoting HTA in Brazil. While the results of the present survey reflect specificities of the Brazilian context, it contributes to the international knowledge on the development of HTA systems. These findings may suit countries with comparable health care systems and encourage the development of their own national HTA strategy [8,54]. Moreover, initiatives for international cooperation to tackle common challenges on HTA implementation should be promoted.

## 5. Conclusions

The implementation and expansion of HTA processes have to be in accordance with the characteristics of the countries’ health care systems and the availability of resources. This survey identified both expectations of a sample of experts regarding the development of HTA and the potential acceptance of selected measures for strengthening HTA in Brazil. HTA structures should increasingly support public health issues and how to appraise evidence on interventions directed at chronic diseases. HTA bodies in Brazil and other emerging countries have to be supplied with appropriate human resources and equipment.

## Figures and Tables

**Figure 1 ijerph-18-11912-f001:**
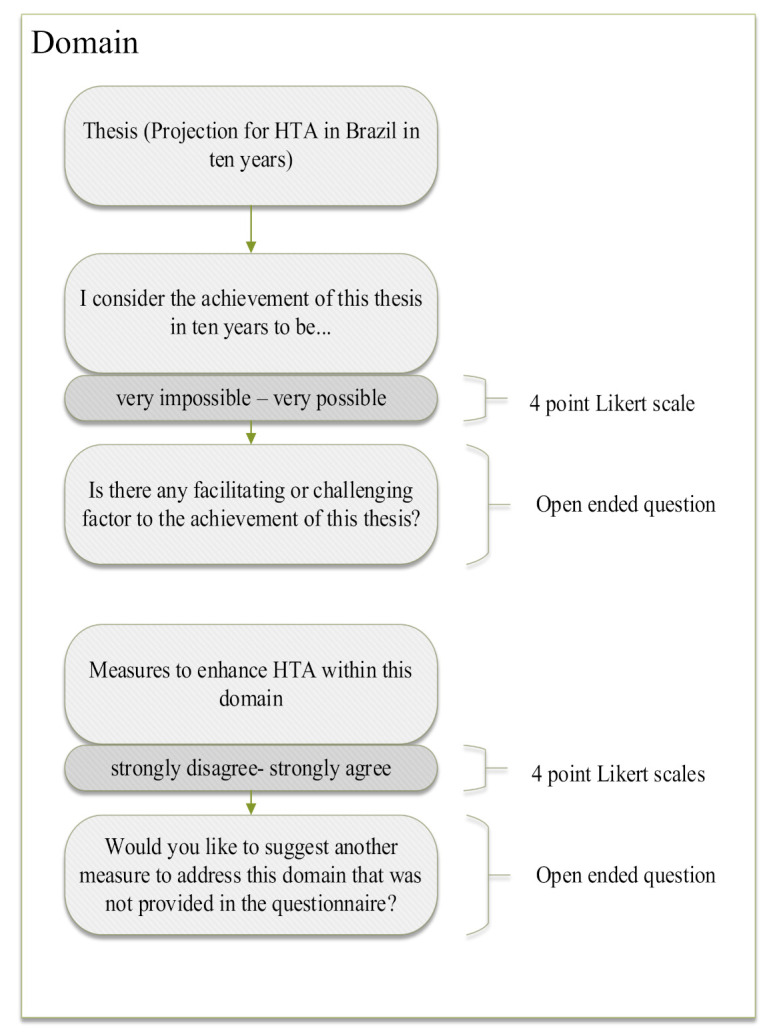
Basic structure of the questionnaire repeated for each domain.

**Table 1 ijerph-18-11912-t001:** Participants.

Groups	Experts Invited (*n*)	Total Responses (*n*)	Complete Responses (*n*) ^1^
NATS	70	23	19
Regulatory	39	6	5
Pharmaceutical industry	15	7	6
Total	124	36	30

Abbreviations: NATS: nuclei for health technology assessment. ^1^ Included in final analysis.

**Table 2 ijerph-18-11912-t002:** Expectation of experts with regard to the theses and respective challenges according to the domains.

Domain	Theses	Expectation of Experts		Challenges to the Achievement of the Thesis
	In 10 Years (Nr.)	Mode (%)	Mean (SD)	
Capacity building	Brazil will have sufficient adequately trained personnel to understand, implement and conduct HTA studies (1)	Possible (83%)	3.0 (0.41)	Lack of funding (8 experts) Education is not a political priority (8 experts) Students without basic skills (i.e., evidence-based medicine, English language) to receive advanced training on HTA (7 experts)
Public participation	Plain public involvement will be guaranteed without compromising the technical quality of the process (2)	Possible (50%)	2.6 (0.56)	Patient representatives or collaborative networks exist for a few groups of diseases (6 experts). Difficulty to identify legitimate patient representatives (2 experts) Conflict of interest of patient representatives funded by the industry (8 experts) Lack of public awareness (3 experts)
Role of cost-effectiveness analysis (CEA)	A CEA will be required in the HTA process to obtain coverage/reimbursement of new technologies in the benefits catalog (3)	Possible (87%)	2.9 (0.36)	Lack of funding for research (5 experts) Need for capacity building on health economics (5 experts) Need for epidemiological and utility data (3 experts)
Institutional framework	A timeframe of three months to complete the HTA will be mandatory (4.a)	Possible (53%)	2.6 (0.66)	Lack of human resources to comply with the deadlines (9 experts) Potential impact on the quality of the reports (2 experts)
	The HTA appraisal process for public and private institutions will be merged (4.b)	Impossible (53%)	2.3 (0.69)	Private and public insurances in Brazil operate from different perspectives (7 experts)
Scope of HTA	The scope of HTA will be restricted to new medical technologies with high added value (e.g., biologics, biosimilars, combination products, devices, oncologic therapy, among others) and potentially high budgetary impact (5)	Possible (67%)	3.1 (0.57)	Difficult to foresee which technology will have a low budgetary impact (1 expert)
Methodology of HTA	The conduction of HTA studies will have high-quality methodology (6)	Possible (70%)	2.9 (0.54)	Lack of funding (5 experts) Lack of reliable epidemiological data on the Brazilian population (5 experts) Need for capacity building (3 experts)
HTA as basis for Jurisdiction	Judicial decisions on individual right-to-health lawsuits concerning the coverage of medical innovative technologies will use CONITEC reports as basis for jurisdiction (7)	Possible (63%)	2.8 (0.58)	Need for capacity building (3 experts) Mistrust between the Judiciary and technical advisors (3 experts)

Abbreviations. HTA: health technology assessment; CONITEC: National Commission for incorporation of Technologies; NATS: nuclei for HTA; CEA: cost-effectiveness analysis; ANS: National Agency for Supplementary Healthcare; SD: standard deviation.

**Table 3 ijerph-18-11912-t003:** Elicited measures to promote HTA in the Brazilian health care system according to the domains.

Domain	Measures to Promote HTA in Brazil according to the Domain [Mean Value (SD)]
Capacity building	Project-based training [3.7 (0.47)], permanent university-based graduate [3.2 (0.91)] and post-graduation programs [3.9 (0.34)]; Courses offered by private universities [3.3 (0.51)], public universities [3.8 (0.42)], international collaborations [3.7 (0.44)] or public–private partnerships [3.7 (0.54)]; In-class traditional courses [3.3 (0.57)], online courses [3.0 (0.75)] or combination of in-class and online courses [3.6 (0.67)].
Public participation	Standardized advertising to the public to stimulate participation in public consultations [3.4 (0.72)]; Mandatory voting seat for patients’ representatives in the Plenaries of CONITEC [2.9 (0.87)]; Education for patients’ representatives on HTA to support participation in public consultations [3.4 (0.75)].
Role of cost-effectiveness analysis (CEA)	Mandatory CEAs for inclusion in the benefits catalog (reinforced by law) [2.9 (0.81)]; Self-binding from the pharmaceutical industry on producing CEAs [3.1 (0.70)]; Responsibility of conduction of CEA by Independent public universities staff [3.6 (0.50)], independent private universities staff [3.2 (0.64)], or an internal capacitated commission of CONITEC [3.1 (0.65)]; CET as reference for the interpretation of the results of the CEA [3.0 (0.80)]; CET as a standard value to be developed by Brazilian researchers, based on the budget available for health in Brazil (or willingness to pay) [3.2 (0.90)]; CET applied as one criterion among others (including ethical, evidence of medical benefit, health care priority, practicability, etc.) [3.6 (0.48)]; Co-financing by the proponent (third parties) and the government would be appropriate and sustainable for encouraging cost-effectiveness studies.^a^
Institutional framework	Reform of the specific legislation to ensure timeliness of reports (3 months) [3.0 (0.75)]; Expansion of the capacity of the NATS [3.4 (0.76)] and of the CONITEC [3.5 (0.67)]; Increase of participation of the ANS in the plenaries of CONITEC [3.0 (0.77)]; Health professionals other than physicians should be represented in decision-making bodies ^a^; Regional representation with regard to ethnical characteristics ^a^; To finance full-time researchers to work at the NATS ^a^.
Scope of HTA	Simplified appraisal procedure to medical technologies with low budget impact (e.g., over-the-counter drugs, organizational and informational protocols, among others) [3.2 (0.78)]; Alternative criteria to prioritize the scope of HTA: ○ Public health needs [3.6 (0.70)], frequent right to health lawsuits concerning a specific technology [3.2 (0.70)], unmet medical needs [3.3 (0.70)], technologies not previously evaluated by other countries [3.0 (0.80)], high therapeutic value of the technology [3.3 (0.77)]. Prioritize technologies for primary health care ^a^.
Methodology of HTA	Expansion of the capacity and training of the CONITEC [3.5 (0.62)] and of the NATS [3.6 (0.55)]; Revision and periodical update of the methodological guidelines [3.6 (0.48)]; Provision of financial support to independent researchers that dedicate themselves to either from public universities HTA [3.5 (0.62)] or from private universities [3.1 (0.70)]; Expansion of the capacity of data collection [3.7 (0.47)] and amelioration of the accessibility of “DATASUS” [3.7 (0.62)]; Implementation of mandatory integrated electronic databases in the main hospitals [3.5 (0.62)] and in centers of the “Family Health Program” [3.5 (0.67)].
HTA as basis for Jurisdiction	Establishment of expert groups to advise the law courts regarding HTA [3.6 (0.66)]; Implementation of early awareness system (or horizon scanning) to identify innovative medical technologies with the potential to become a target of lawsuits [3.4 (0.55)]; Mandatory HTA appraisal by CONITEC after authorization to enter the market is granted by ANVISA, to ensure timely reports [3.2 (0.78)]; Invest in the expansion of the NATS JUS ^a^.

^a^ Measures suggested as free text, therefore not appraised by the group. Abbreviations: HTA: health technology assessment; CONITEC: National Commission for incorporation of Technologies; NATS: nuclei for HTA; CEA: cost-effectiveness analysis; CET: cost-effectiveness Threshold; ANS: National Agency for Supplementary Healthcare; ANVISA: National Health Surveillance Agency; DATASUS: Information Technology Department of the National Healthcare system; NATS JUS: NATS dedicated entirely to assisting the judiciary with regard to technical issues on HTA; SD: standard deviation.

## Data Availability

All data generated during the conduction of this study are reported in the main manuscript and Appendix A.

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
