# Peer review of "Expectations for the Development of Health Technology Assessment in Brazil"

_ijerph, 2021, doi:10.3390/ijerph182211912_

Round 1
Reviewer 1 Report
First of all, congratulations for the work done, for your time and dedication.
Second, I will make some recommendations or suggestions to improve the manuscript.
The implementation of health technology assessment in emerging countries dependent on the health system and the need for public health care. Evaluating the expectations of the experts for its development in Brazil. A questionnaire has been developed that includes the bases for the development of health technology. Thirty experts responded and suggested measures to equip the system to assess health technologies.
A questionnaire was developed on seven domains that reflect the current debates about HTA in Brazil.
In development of the method for the elaboration of the questionnaire it is well detailed.
Even so, I have some doubts, to complement in your discussion:
-What does this work mean for society, what benefit or implications does this work have?
-What does this work mean for the advancement at the theoretical level of the scientific community or theoreticians.
-What future lines of research or gaps have you seen on this topic and do you see them as future lines of research.
Please incorporate these comments at the end of your discussion, and my sincere congratulations on the preparation of this work.
An affectionate greeting,
Author Response
Thank you for the overall positive review of the manuscript. We included the suggested points on lines 317-318 and lines 322-330 at the end of the discussion as suggested.
Reviewer 2 Report
Overall, an interesting look at HTA in a local health service area. The paper is generally well written, some improvements could be made in the language particularly in the introduction. For example, Line 29 - past tense? (focused). Some further specific comments below.
Intro: How does HTA apply to the health system globally and locally? Provide some examples of where HTA has been used. How do legal and ethical obligations align with HTA? Some further background information would be beneficial to allow the reader to understand the concept of HTA, its use and its applicability to different health care streams and organisations. Why does it need to be expanded?
Methods: Did you use a framework for your scoping review (e.g. PRISMA)? Table 1A would be better included in your methods section. The questionnaire is comprehensive and covers the domains well.
Results: You have described a scoping review but have not presented any of the findings of this review. Is this review published elsewhere? Or can it be included in this paper? It would be beneficial to know how many articles you utilised to develop your questionnaire. For example, how many articles were aligned with each domain that you identified? A table outlining this would be helpful in understanding the importance of each barrier and domain. In Table 2 you use the wording "possible" but in the questionnaire it is "probable". It would be beneficial to keep this consistent. Otherwise, the results are well presented.
Discussion: The discussion brings together your results well. The limitations have been addressed clearly and the conclusions align well with the aims of the study.
Author Response
Thank you for reviewing the manuscript and for the constructive comments. To facilitate the review process, the comments were copied here and responses to each comment are highlighted in red. All revisions to the manuscript are marked up using the Track Changes function. We stay at your disposal for further clarifications.
Overall, an interesting look at HTA in a local health service area. The paper is generally well written, some improvements could be made in the language particularly in the introduction. For example, Line 29 - past tense? (focused).
Line 29, now line 35 – corrected to past tense.
Intro: How does HTA apply to the health system globally and locally? Provide some examples of where HTA has been used. How do legal and ethical obligations align with HTA? Some further background information would be beneficial to allow the reader to understand the concept of HTA, its use and its applicability to different health care streams and organisations. Why does it need to be expanded?
We addressed these points on lines 28-33 and lines 51-53 of the introduction.
Methods: Did you use a framework for your scoping review (e.g. PRISMA)?
The scoping review was a preliminary step for the preparation of the questionnaire (which was the focus of the analysis). Although a specific framework such as PRISMA for systematic reviews was not used, we followed an iterative stage process as proposed by Arksey and O’Malley (2005). More details on search strategy, study selection and data charting were added to increase transparency. (lines 72-78)
Table 1A would be better included in your methods section. The questionnaire is comprehensive and covers the domains well.
Table 1A was kept on the supplementary material; but we now added a short description of the search strategy on lines 86-88.
Results: You have described a scoping review but have not presented any of the findings of this review. Is this review published elsewhere? Or can it be included in this paper? It would be beneficial to know how many articles you utilised to develop your questionnaire. For example, how many articles were aligned with each domain that you identified? A table outlining this would be helpful in understanding the importance of each barrier and domain.
We added to the methods section a description of the studies included in the scoping review (lines 90-98).
In Table 2 you use the wording "possible" but in the questionnaire it is "probable". It would be beneficial to keep this consistent. Otherwise, the results are well presented.
We adjusted the questionnaire on the supplementary material to keep it consistent.
Discussion: The discussion brings together your results well. The limitations have been addressed clearly and the conclusions align well with the aims of the study.
Round 2
Reviewer 2 Report
Thank you for your reply. You have addressed the comments provided very well. Your manuscript is much improved. Well done.